# DPP: Deep predictor for price movement from candlestick charts

**Chih-Chieh Hung** [1] *, **Ying-Ju Chen** [2]

**1** Department of Management Information Systems, National Chung Hsing University, Taichung City, Taiwan, **2** Department of Mathematics, University of Dayton, Dayton, Ohio, United States of America

* smalloshin@nchu.edu.tw

## Abstract

Forecasting the stock market prices is complicated and challenging since the price movement is affected by many factors such as releasing market news about earnings and profits, international and domestic economic situation, political events, monetary policy, major abrupt affairs, etc. In this work, a novel framework: deep predictor for price movement (DPP) using candlestick charts in the stock historical data is proposed. This framework comprises three steps: 1. decomposing a given candlestick chart into sub-charts; 2. using CNN-autoencoder to acquire the best representation of sub-charts; 3. applying RNN to predict the price movements from a collection of sub-chart representations. An extensive study is operated to assess the performance of the DPP based models using the trading data of Taiwan Stock Exchange Capitalization Weighted Stock Index and a stock market index, Nikkei 225, for the Tokyo Stock Exchange. Three baseline models based on IEM, Prophet, and LSTM approaches are compared with the DPP based models.

## Introduction

The stock market performance is one important indicator for the world economy. It provides an overall insight about the performance of all listed companies in a country. Stock market price forecasting has been grabbing a great amount of attention since the introduction of the first publicly shareholder company: The Dutch East India Company in the 17th century [1] due to 1. it allows investors to maximize the profit in the stock market; 2. it helps large-scale enterprises to acquire small businesses or merge with another corporation at a good time point; 3. it aids investors, banks, and insurance companies to avoid potential risks on large fluctuation of stock prices.

However, such forecasting problem is complicated and challenging since the price movement is affected by many factors such as releasing market news about earnings and profits, international and domestic economic situation, political events, monetary policy, major abrupt affairs, etc. There are two main branches of study in the literature: Fundamental Analysis and Technical Analysis. Though the major purposes of these two methods are the same, the focuses of analytical tools are quiet different. The former focuses on both past and present data corresponding to economic reports, market news, and industry statistics, while the latter looks at

**Funding:** We would like to express that there is no funding supported for this research work.

**Competing interests:** The authors have declared that no competing interests exist.

statistical trends such as price movements from past data with chart analysis [2]. In this article, we propose a deep learning framework applied in candlestick pattern analysis which is one of the popular tools of chart analysis in assistance of forecasting stock market price [3–6].

*Candlestick charts* have been the most popular visualized tools that summarize daily price, currency movements, volumes, or values of technical indices. Fig 1 shows an example of a 20-day candlestick chart and Fig 2 shows the essential components of candlesticks.

A candlestick chart consists of sticks for a specific period of time and each stick displays the range between opening and closing traded prices (body) and the highest and lowest traded prices for one day trading data. In addition, one should note that the use of colors for candlestick charts could be different depending on countries. Due to the simple structure of candlestick charts, they are straightforward and easy to read. It is easy to recognize visual trends from candlestick charts and use the corresponding information collected to help predict price movements. Although the literature has shown that candlestick pattern analysis is a successful approach in financial forecasting [7–10], predicting the price movements by reading the visual trends from candlestick charts is still challenging. One can argue that nuances are lost in the candlestick charts while candlestick charts simplifies complex analysis behind the scene. In fact, each stick reflects the difference between any two values among open, closing, highest and lowest traded prices and including candlesticks side-by-side would show the deviation between daily traded prices during the period of time. While the trading volume is not included in a candlestick chart, there is no clear evidence in the literature supporting that the trading volume improves the performance of forecasting stock price movements [11–13]. Zhu et al. [13] mentioned that including the information of total market trading volume does not contribute to the improvement of network forecasting since it may also contain any unnecessary or inter-correlated information.

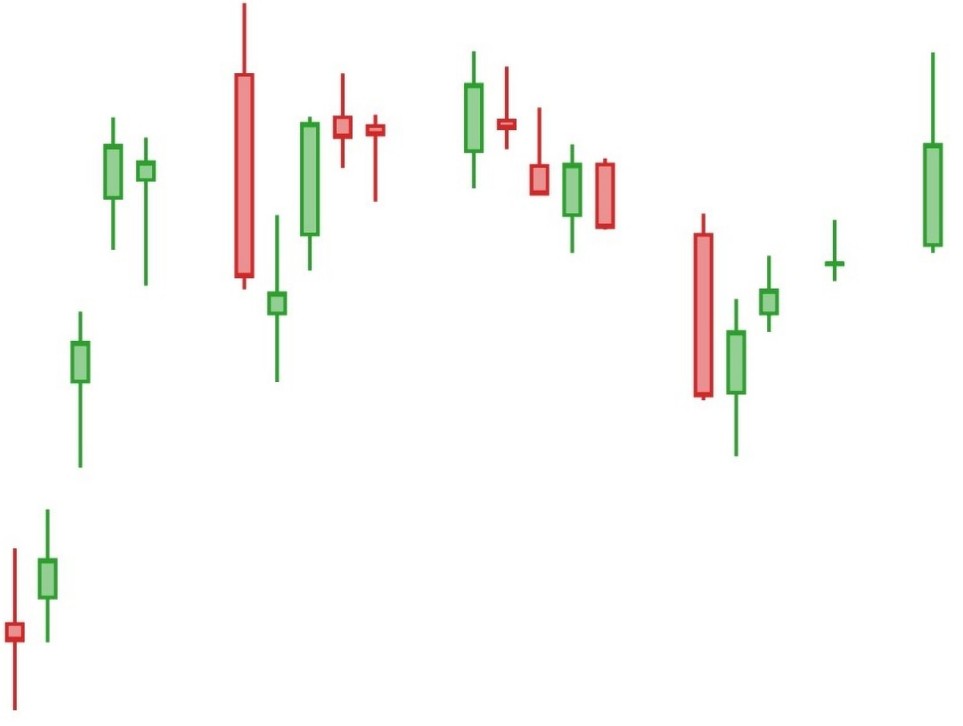

**Fig 1. A 20-day candlestick chart.**

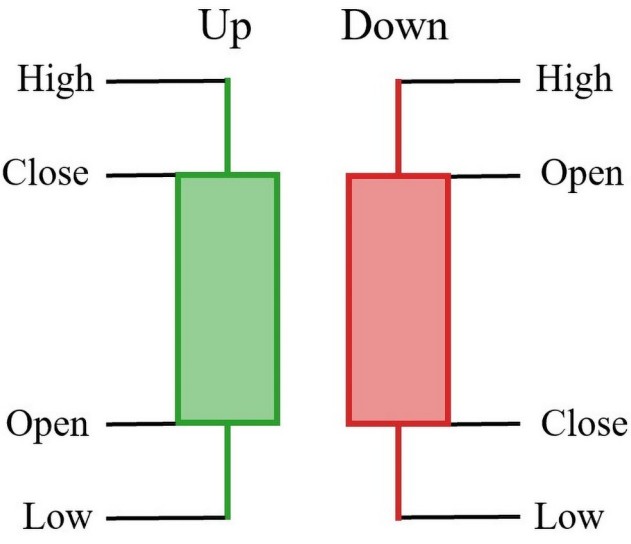

**Fig 2. The basics of candlesticks.**

When using candlestick charts to forecast the price movements, there are some challenges that need to be solved. The first challenge comes from the inclusive of the non-informative blank space in a candlestick chart as the focus should be the informative part of it, i.e., the candlesticks. Thus, the development of a refined pre-processing approach to read candlestick pattern is essential. The second challenge comes from the extraction of representative characteristics from sub-charts since individuals prefer to identify local trends at first glance instead of finding the global pattern. The third challenge is about learning the meta-patterns and series of local patterns in candlestick charts since they usually bring more attentions to humans when studying the price movements. Instead of reading the pre-decided patterns visually, the major contribution of this article is development of a predictive model to find features that are highly associated to the price movements using candlestick charts.

A deep learning framework DPP (*Deep Predictor for Price* Movement) is introduced in this article to predict the price movement of a given day, say $(k + 1)$-th day, by taking a $k$-day candlestick chart as an input. Latane et al. [14] defined the price movement by comparing the price of the $(k + 1)$-day and the price trend of the past $k$-days. Specifically, the price movement is defined to be upward if the price of the $(k + 1)$-th day is larger than $\mu_k + \alpha\sigma_k$ where the $\mu_k$ and $\sigma_k$ refer to the mean and standard variation of the price in the past $k$ days and $\alpha$ is a user-specified parameter; otherwise, the price movement is defined to be downward [14]. For example, given Fig 1 which $k$ is set to be 20, the 20-day candlestick chart, DPP can be used to study the price movement of the 21st day is upward or downward.

The design of DPP is illustrated in Fig 3. It consists of a chart decomposer, a CNN-Autoencoder (abbreviated as CAE) and an RNN with GRU gating mechanism (abbreviated as GRU). CNN stands for Convolution Neural Network, RNN stands for Recurrent Neural Network, and GRU stands for Gated Recurrent Unit. The concept of DPP is to decompose the candlestick chart into sub-charts, extract the graphical features from these sub-charts, and predict the price movement by the time-series of these sub-charts. In the initial step, the chart decomposer splits a given $k$-day candlestick chart into multiple $m$-day sub-charts. The first challenge could be addressed by standardizing the sizes of the bodies and the shadows of candles for all $m$-day sub-charts. Then these sub-charts are adopted to train a *CNN-Autoencoder* in the second step

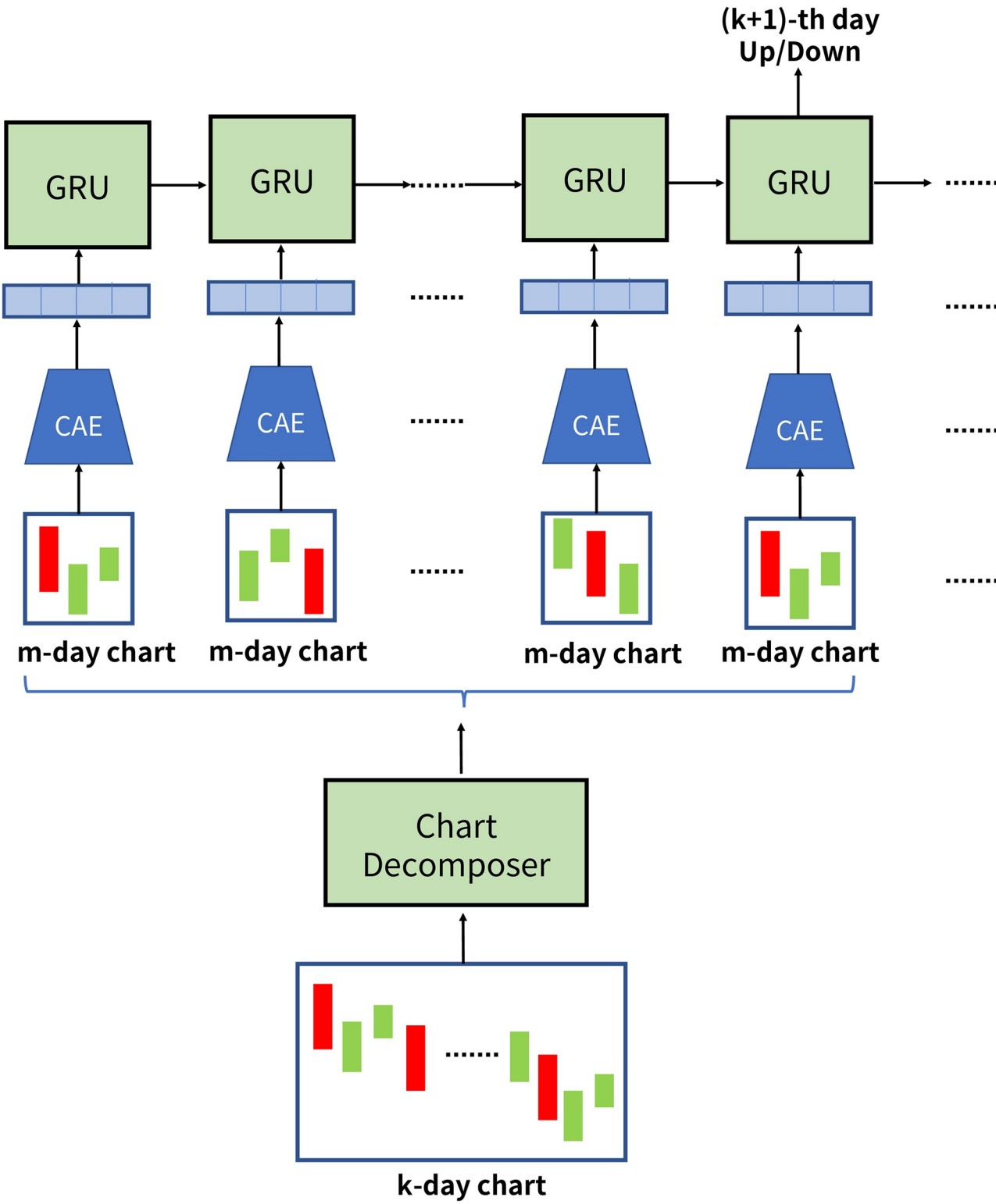

**Fig 3. Overview of the proposed framework.**

which can extract deep features from sub-charts. The second challenge is solved since the features extracted could be the best representative of the sub-charts. In the final step, *RNN with GRU gating mechanism* is used to predict the $(k + 1)$-th day price movement from the sequences of these deep features. Thus, the third challenge is clear up because the relationship between the price movement and the series of the deep features could be learned by RNN.

In order to assess the performance of DPP based models, two datasets are adopted, the trading data of Taiwan Stock Exchange Capitalization Weighted Stock Index and Nikkei 225 which is a stock market index for the Tokyo Stock Exchange, and three baseline approaches, IEM, Prophet, and LSTM, are considered. The result suggests that the overall performance of DPP approach is better than it of baseline approaches.

## Related work

We will discuss some related work in the literature corresponding to time series forecasting of candlestick charts developed on the deep-learning approaches.

### Deep learning approaches for time series forecasting

Forecasting problems in time series have been studied in the field of Machine learning which has brought lots of attention in recent decade. Recurrent Neural Networks (RNN), a variant of neural network, is one of the popular models for solving such types of questions. Besides the flexibility in dealing with varying lengths of sequences, RNN also provides advantage of memorizing the features learned in different positions of sequences as its recursive architecture [15, 16]. These features make RNN heavily-used in time series forecasting tasks, such as natural language processing.

According to the number of inputs and outputs, RNN could be classified into four categories: one-to-one RNN, one-to-many RNN, many-to-one RNN, and many-to-many RNN. For example, many-to-one RNN accepts several inputs and gives one output. Since the problem addressed in this paper is to use many candlestick charts to predict the price movement in the $(k + 1)$-th day, many-to-one RNN is adopted. Apart from the power of RNN, one of the main issues in training RNN is the gradient vanishing [17]. Gradient vanishing refers to the problem that the gradient components of the loss function approach zero which makes the network hard to train. This weakness makes RNN unable to capture long term dependencies in sequences. To overcome this issue, a modification of RNN architecture is needed, such as Long Short-Term Memory (LSTM) [18] and GRU (Gated Recurrent Unit) [19]. LSTM has three gates in each memory cell in RNN: an update gate, an input gate and an output gate. As each gate contains a sigmoid activation functions and pointwise multiplication operation, LSTM can add or remove information to the memory cell. GRU is similar to LSTM but it has two additional gates, update gate and reset gate, and an extra activation function (hyperbolic tangent, tanh) in each memory cell. Different from LSTM, GRU passes the memory content that is seen or used by other units to the next memory cell directly instead of being controlled by gates. Compared to LSTM, GRU takes less training time and needs fewer training data since GRU requires fewer parameters when GRU could achieve a similar performance according to previous empirical tests [20]. Therefore, GRU is selected to predict the price movement in our study.

### Analysis of candlestick charts

There have been lots of studies in forecasting the price tendency through candlestick charts. The candlestick-chart-formed data and pre-defined patterns are adopted to assess the performance of hybrid stock market forecasting models in Takenori Kamo et al. [21]. Karsten

Martiny [22] introduced the tree-based pattern-search method in aims of discovering essential candlestick patterns and further predicting future price movements.

The original study that attempts to mimic how human traders make trading decisions through reading numerical candlestick-chart-formed data is from [23, 24], which conducted hybrid machine learning models to study timing of the stock markets. Reading candlestick charts allows traders to understand impacts of trends called *visual features rather than reading numerical raw data directly*. Some recent investigations manifest that analyzing visual characteristics of candlestick charts to predict stock market is still a hot topic, for examples, Hu et al. [8] summarize the historical financial data as images using candlestick charts and adopt the convolutional autoencoder for feature learning from the image data, Fengqian and Chao [25] apply K-line theory to characterize candlesticks as a generalization of price movements over a time period and then propose the deep reinforcement learning based system to reach adaptive control in the unknown environment, and Ananthi and Vijayakumar [26] proposed a system that generates signals on the candlesticks to predict market price movement by using regression and candlestick pattern detection. Furthermore, a novel variation of conventional candlesticks, RGBSticks, is introduced in [27] to predict the daily stock price of a company by using an autoencoder based on deep neural network.

Although the approaches above appear to be comparable to our work in this paper, one single settled time interval would not be considered by traders and long time period candlestick charts will be divided into several sub-charts for analysis.

This study is an extended version of our previous work [28, 29]. To our best knowledge, our previous work is the first study that uses $k$-day candlestick charts to predict the price movement which is defined as the difference of the close prices of $k$-th day and the $(k + 1)$-th day [28, 29]. In this extended version, some improvements are made which can distinguish this paper from our previous study. First of all, we develop a chart-decomposer to crop sub-charts from a $k$-day candlestick chart, rather than using raw data to generate sub-charts, which is consistent with the scene of this paper: reading the candlestick to predict price movement. Secondly, the definition of price movement in this paper takes the mean and variance of prices of $k$-day history into account, rather than simply uses the difference of prices of the $k$-th day and the $(k + 1)$-day, which can better represent the trend of the price movement. Thirdly, the predictor of the price movement is based on GRU, rather than 1D-CNN, which is more intuitive and brings higher accuracy in nested cross-validation. At last, nested cross-validation, rather than simply k-fold cross-validation, is used to validate models in the experiments in this paper, which could better capture the nature of temporal dependency in time-series data and reflect more reliable experimental results. In brief, we believe this extended version has made significant contributions compared to previous work.

## Deep predictor for price movement (DPP)

In this section, the proposed framework DPP is introduced with the detailed procedures in each stage.

The target of prediction in this paper is the price movement. Let $C_d$ be the close price of $d$-th day and $d \geq k + 1$, the price movement of the $d$-th day is defined as follows:

$$target(d) = \begin{cases} 1, & C_d > \mu + \alpha \times \sigma \\ 0, & otherwise \end{cases} \tag{1}$$

where $\mu = \frac{\sum_{t=d-k}^{d-1} C_t}{k}$, $\sigma = \sqrt{\frac{\sum_{t=d-k}^{d-1} (C_t - \mu)^2}{k-1}}$, and 0 and 1 represent downward and upward of price movement, respectively. If the close price of $d$-th day is greater than $\alpha$ standard deviations above the mean of past $k$ days' close prices, then the price movement is defined as upward. Otherwise, the price movement would be downward.

Note that the price movement here borrows the concept of Bollinger Bands [30] which is a widely-used technical analysis tool to give a relative definition of high and low prices of a market, allowing traders to vary the response of the chart to the magnitude and frequency of price changes. Bollinger Bands consists of an $d$-period moving average (same as $\mu$ here), an upper and lower band at $\alpha$ times an $d$-period standard deviation (same as $\sigma$ here) above and below the moving average, respectively. To model the price prediction into a binary classification problem, our definition of the price movement can be viewed as a simplified Bollinger Bands.

In this paper, the default value of $k$ is set to be 20 and that of $\alpha$ is set to be 1. The goal of this work is to maximize the accuracy of the predicted price movement over the actual price movement.

## Chart decomposer

In the first state, we decompose a given $k$-day candlestick chart into several $m$-day sub-charts. The specification of a sub-chart needs to be settled carefully since these sub-charts will be utilized to extricate features within the afterward stages.

The development of sub-charts in this paper takes after two standards: 1. produced sub-charts must comprise $m$ sticks and the corresponding shadow and body of the sticks are required to be plainly visible and 2. the dimension of produced sub-charts need to be as little as possible in aim of reducing the area of non-informative blank space. In this study, we expect to have high-quality candlestick charts and they are well-formed with consistent width. To decompose a given candlestick chart into sub-charts, the unnecessary part of the candlestick chart is firstly cropped by deriving a minimum bounding rectangle (MBR) of non-white pixels which have the height $\ell$ and the width $w$. Suppose that the candlestick chart is well-formatted. We can divide the derived MBR into 20 rectangles with height $\ell$ and width $\frac{w}{20}$. Then, we scan the image of each rectangle line by line to identify the candlestick. Finally, given any consecutive $m$ candlesticks, we can generate an image with specific length and width.

It requires in general at least 3 candlesticks to identify patterns such that three line strikes and three black crows. Thus, we use 3-day sub-charts in our study and set $m$ to be 3 as the default value. Fig 4 gives an example of a 3-day sub-chart. In this paper, a 3-day sub-chart is a squared, grey-scaled image with the length 48 pixels where a while-colored stick means the price rises, a grey-colored stick means the price falls, and the background is black. For each stick, the width of shadow is 1 pixel and the width of body is 10 pixels.

## CNN-autoencoder

We use a CNN-autoencoder (CAE) to extract the deep features from these sub-charts after acquiring a few $m$-day sub-charts in the first stage. CAE contains two parts: encoder and decoder which learn a representation from the data. For the encoding procedure, each sub-chart is converted to a deep feature using several convolutions and pooling layers in CNN. Then in the decoding procedure the sub-charts are reconstructed by striding transposed convolutions. The encoding and decoding procedures of CAE aim to obtain the best representation of the input-charts by recovering these sub-charts from the output images obtained from deep features.

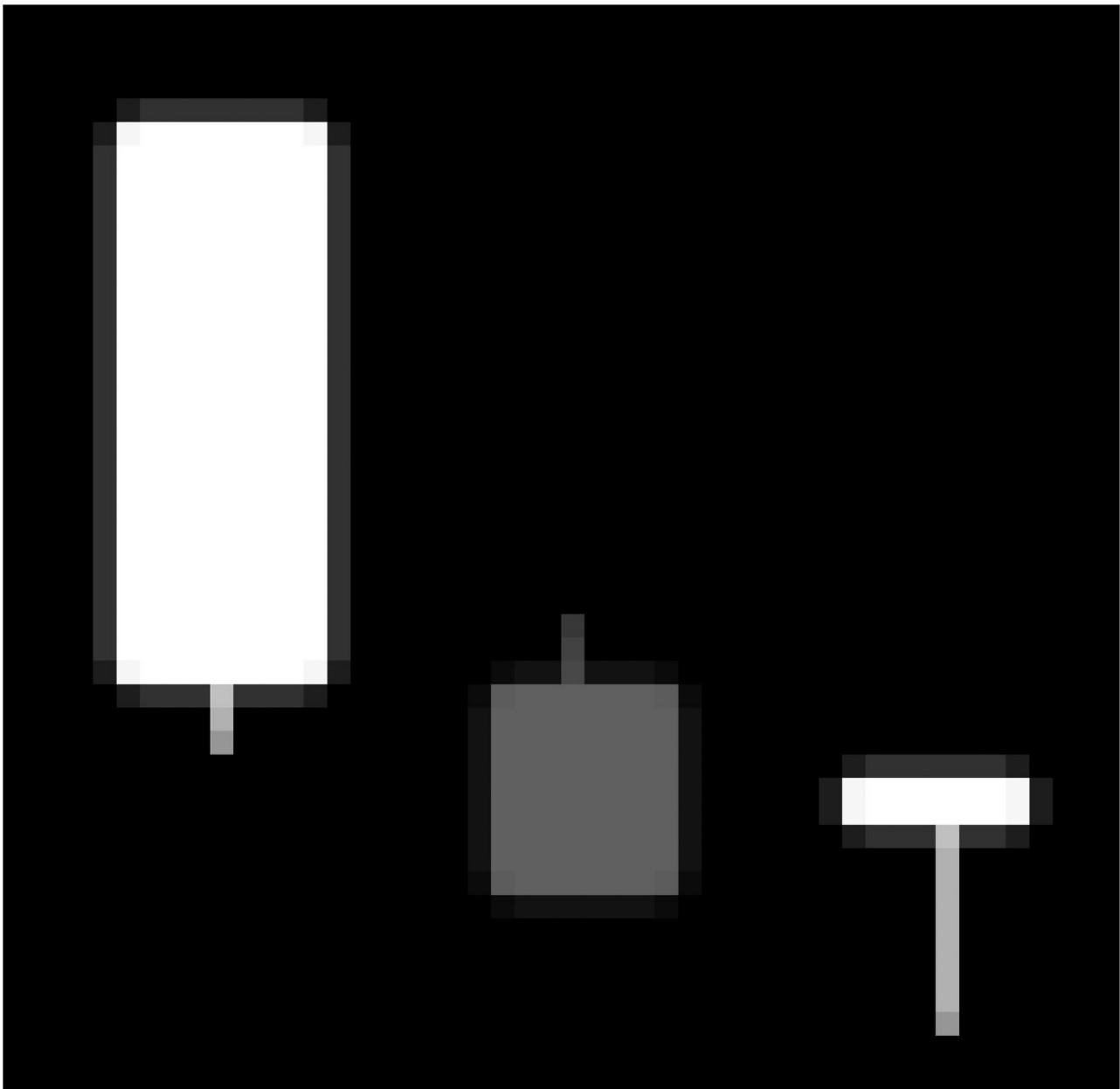

**Fig 4. An illustrative example of a 3-day sub-chart.**

The structure and corresponding parameters of the CAE in the study are illustrated in Fig 5. First, the input dimension of CAE is $48 \times 48 \times 1$ because of the use of a 48 pixels gray-scaled image of 3-day sub-chart. Second, there are 3 components in the encoding process: a 2D convolution layer (Conv-2D), a 2D max pooling layer (MaxPooling-2D) with ReLu as its activation function, and the parameters of each convolution layer are appeared in its bottom. Third, as the network becomes deeper, the number of channels for the conventional CNN increases and its dimension of intermediate images diminishes. Fourth, the deep feature could be achieved to have the dimension of $3 \times 3 \times 64$ via the average pooling in the 8th layer. The

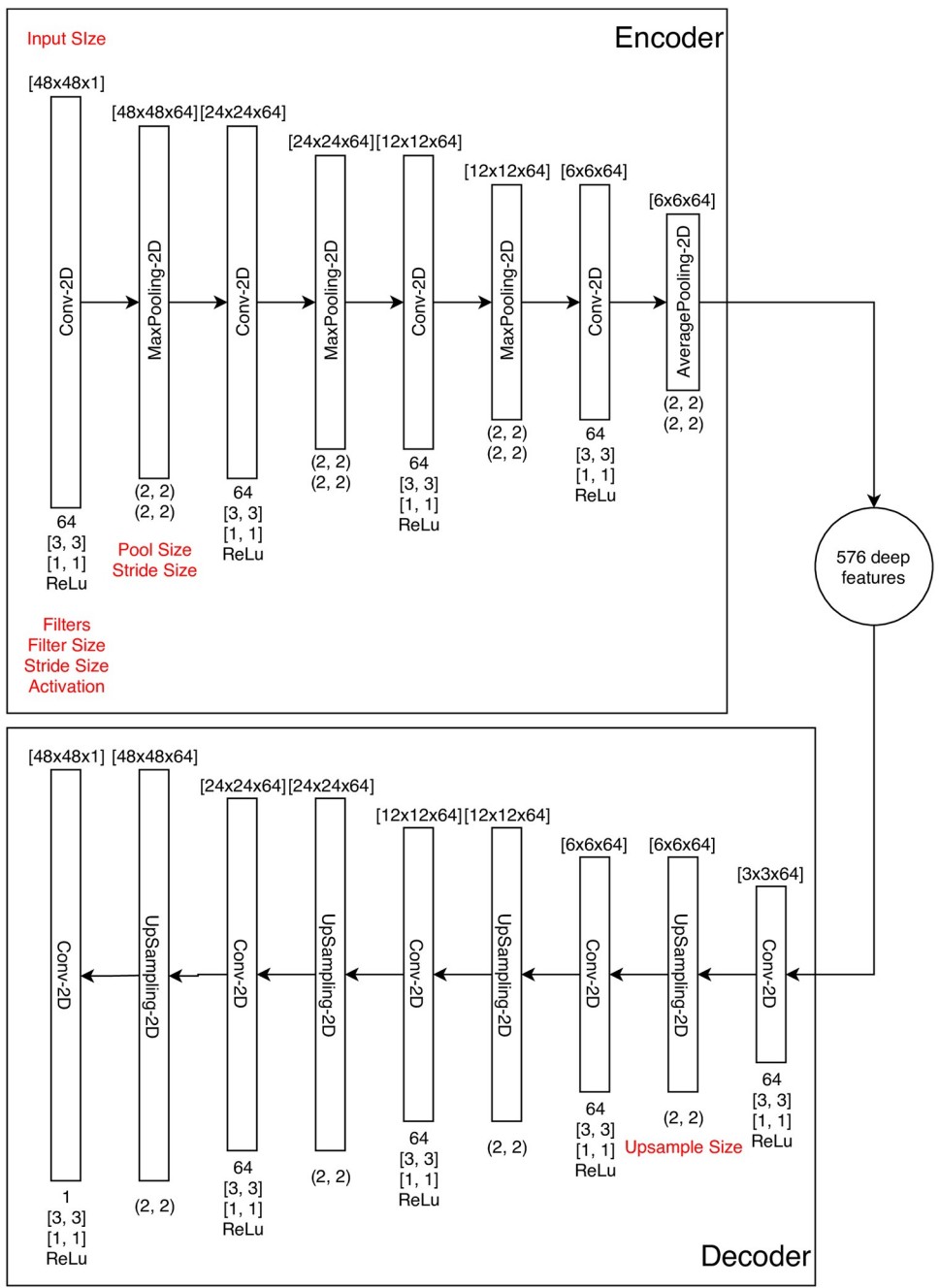

**Fig 5. CNN-autoencoder.**

decoding process is proposed to be the reverse of the encoding process. Finally, the dimension of the decoder is $48 \times 48 \times 1$. Here, we note that the loss function of the CAE selected is Binary Cross Entropy which is used to assess the difference between the original 3-day sub-charts (input) and the output images. Besides, the additional manual checking for the output images is required as the high likeness between the original sub-charts and the output images is not always guaranteed through minimizing the loss function. Besides, the additional manual

checking for the output images is required as the high likeness between the original sub-charts and the output images is not promised through minimizing the loss function.

## RNN with GRU gating mechanism

In the third stage of the proposed framework, GRU is used to predict the price movement after transforming sub-charts into deep feature vectors. To achieve this goal, given a $k$-day candlestick chart, there would be $k - m + 1$ deep feature vectors obtained by transforming $k - m + 1$ sub-charts through CAE. Since these sub-charts are time-ordered, a sequence of the corresponding $k - m + 1$ deep feature vectors is taken as the input of the GRU and the output of the GRU is the predicted price movement (up/down).

As one of the famous deep learning mechanism for time-series classification problems, RNN faces the gradient vanishing problem so that it usually can not achieve satisfiable performance especially when the input is a long sequence. Many gating mechanisms are invented to overcome the gradient vanishing problem. The GRU, as the state-of-art gating mechanism, is like a long short-term memory (LSTM) with a forget gate but has fewer parameters than LSTM [19] has. Since GRU is reported to be able to achieve a similar performance but has better training efficiency compared to LSTM, GRU is selected to forecast the price movement in this paper.

Fig 6 displays the implementation of GRU in this paper where the red-colored values are the possible values for hyper-parameters. For the demonstration purpose, $k$ is chosen to be 20

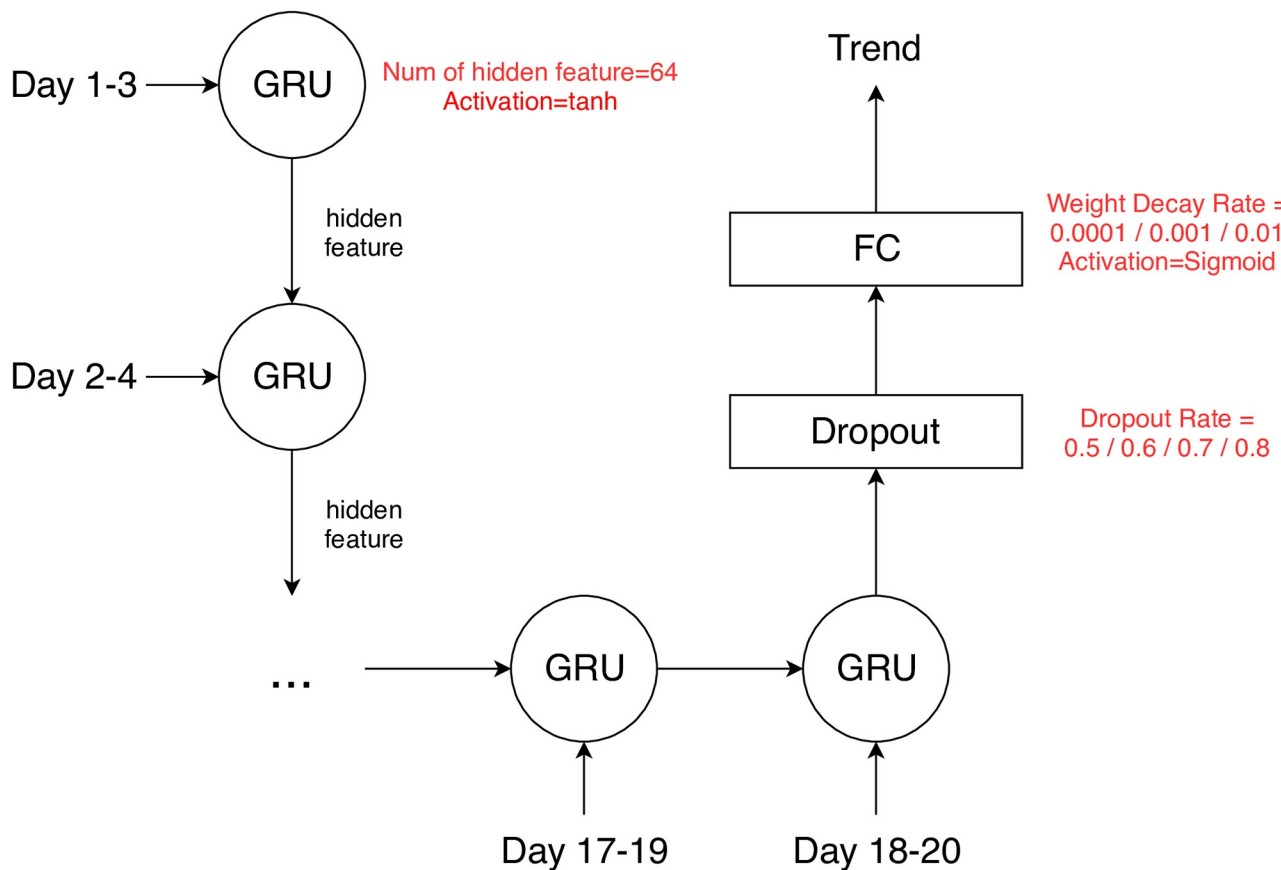

**Fig 6. RNN with GRU gating mechanism.**

and *m* is chosen to be 3. A 20-day candlestick chart is transformed into 18 deep feature vectors by CAE such that the size of each deep feature vector is $3 \times 3 \times 64 = 576$ pixels. The RNN is composed of 1-layer GRU which takes 18 sequential deep features from CAE and then outputs a deep feature vector, and a fully connected layer is followed to classify the price movement of the 21st day. Here, the fully connected layer is composed of one input layer whose size is the same as the number of hidden features in GRU, and one output layer whose size is 1 neuron. The dropout and weight normalization in fully-connected layer are also used to prevent the fully-connected-layered classifier from overfitting problem [31, 32]. The final output is a binary value, telling up or down. Note that there are several hyper-parameters to be tuned by cross validation. The performance of different settings for hyper-parameters used here will be discussed in latter section.

## Experiment results

In this section, the performance of the proposed method DPP will be assessed in comparison with three baseline methods. The setting of the experiments, the preparation of data, the assessment of model performance as well as the sensitivity analysis for hyper parameters in DPP will be covered.

### Settings and datasets

**Baselines.**   We will compare the proposed approach DPP with the three baselines *IEM*, *Prophet*, and *LSTM*. The overview of the baseline models is provided as follows:

- IEM [33] is an index-based approach by taking 10 traditional indices into account, which are simple moving average (SMA), weighted moving average, momentum, stochastic K, stochastic D, RSI, MACD, Lary William's R, A/D(Accumulation/Distribution) Oscillator, and CCI. Given a historical trading data set, IEM firstly uses 10 indices to determine the price movement, according to human-specified thresholds set by domain experts, and then it trains 1 classifier to ensemble the results from 10 decisions to generate the prediction of the price movement. In the following experiment, the classifier in IEM model is chosen to be Support Vector Machine (SVM) as it achieves the highest accuracy in nested cross-validation based on the data sets selected.

- Prophet [34] is an additive model to predict time-series data. Specifically, a time series model can be decomposed into three additive components: $y(t) = g(t) + s(t) + h(t) + \epsilon$ where $y(t)$ is the value of a time series at time t, $g(t)$ is a trend function that models non-periodic changes, $s(t)$ is a seasonality function that models the periodic changes, $h(t)$ is the function which reflects the effect of holiday, and $\epsilon$ is the error. In this experiment, we use the open-sourced implementation of Prophet model [34] by setting $y(t)$ as the close price on day $t$, $g(t)$ as the piecewise logistic growth model, $s(t)$ as Fourier series, and $h(t)$ as the normal distribution with zero mean.

- LSTM (Long Short Term Memory networks) [35] is a variant of recurrent neural networks that can learn long-term dependencies. Many existing studies for stock price prediction take LSTM as their baseline model. Following the suggestion in [35], in our study the model LSTM takes 10 features as inputs, including the raw data (Open, High, Low, Close prices, Volume), and indices (EMA12, EMA25, MACD, BollingerUp and Bollinger Down), to predict the close price of the next day. Then, we apply Eq 1 to generate the prediction target (i.e., upward and downward).

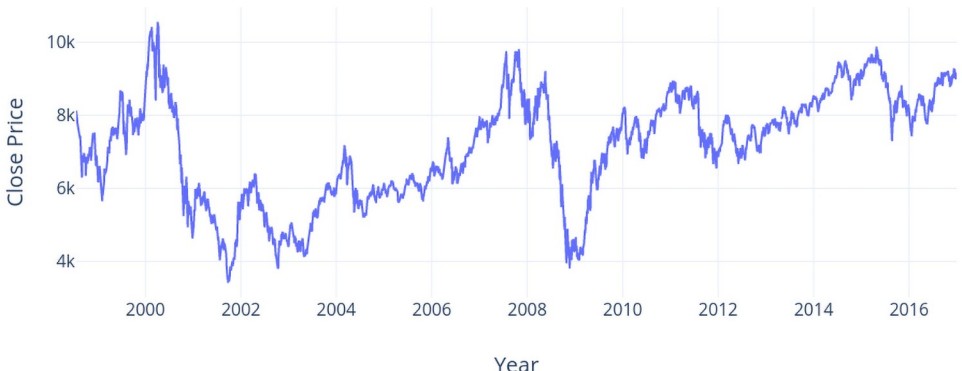

**Fig 7. Close prices over years (TX dataset).**

**Datasets.** Two datasets are used to explore the performance of DPP based models and baseline models. The first dataset (TX dataset) is composed of four futures: TX (TAIEX Futures), MTX (Mini-TAIEX Futures), TE (Electronic Sector Index Futures), and TF (Finance Sector Index Futures). These futures are issued by the Taiwan Futures Exchange which offers futures and options on major Taiwan stock indices, government bond futures, equity options and 30-day CP interest rate futures. As TX is one of the most active domestic investment products in Taiwan, we predict the price movement of TX in this experiment while other futures are used in model training. The interval of TX dataset is from July 21th, 1998 to December 27th, 2016, i.e., 4,625 trading days. Since TX dataset includes four futures, TX dataset includes trading data for 18,500 trading days totally. The close price of TX dataset in the interval is shown in Fig 7.

The second dataset (NI225 dataset) is Nikkei 225, which is a stock market index for the Tokyo Stock Exchange. NI225 measures the performance of 225 large, publicly owned companies in Japan from a wide array of industry sectors. The choice of this data set is based on the similarities between the economies and cultures of Taiwan and Japan as well as sharing the same color theme in the candlesticks (Red for up and Green for down). The time interval of NI225 dataset is from January 5th, 2001 to November 30th, 2020, i.e., 4,983 trading days in total. The close price of NI225 dataset in the interval is shown in Fig 8.

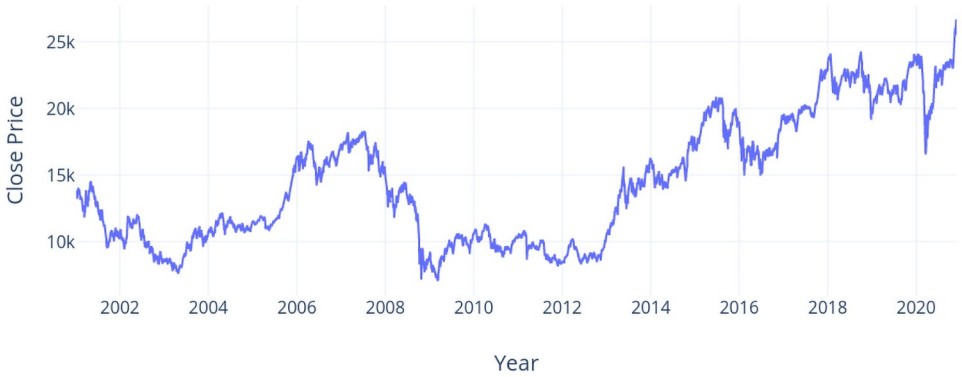

**Fig 8. Close prices over years (NI225 dataset).**

We note that there does not exist class imbalance issues in both datasets. The proportions of prices moving up and moving down are about the same in all merchandises.

**Metrics.**  In order to assess the performance of models, five performance measures are used: accuracy, precision, recall, true positive rate and false positive rate. Because forecasting price movements can be modeled as a dichotomous classification problem, four values *FP*, *TN*, *FN*, and *TP* can be calculated directly given a classification model, where *FP* is the number of false positives, *TN* is the number of true negatives, *FN* is the number of false negatives and *TP* is the number of true positives. The definition of the five performance metrics can be calculated directly as follows:

- accuracy: $\frac{TP+TN}{TP+TN+FP+FN}$

- precision: $\frac{TP}{TP+FP}$

- recall: $\frac{TP}{TP+FN}$

- true positive rate: $\frac{TP}{TP+FN}$

- false positive rate: $\frac{FP}{FP+TN}$

**Validation.**  Since data from financial indices are time series, nested cross validation is used to validate the model performance [36]. Due to the nature of time series data, the training data and test data should be split chronologically. This aims to simulate the situation that we stand in the present and forecast the future so that the data after "the present" should not be used for fitting the model.

In our experiment, the TX data from 1998 to 2007 are first considered. The TX data in this time period are split into three sets: the training data consist of the data from 1998 to 2005, the data for 2006 are used for the validation of models, and the 2007 data are served as test data. After that, the considered time period keeps extended by one year until 2016 and the data in the corresponding time period are split in the same way as described above. Similarly, for the NI225 dataset, we first consider data from 2001 to 2009. The training data consist of the data from 2001 to 2009, the validation data is the data of 2010, the testing data is the 2011 data. The considered time period keeps extended by one year until 2019. Features used in each model could be derived from merchandises data and response variable is the price movement for a given index on a selected date. The generation of features in different models may need data from various number of trading dates. For example, suppose the duration of trading dates in the TX data are from 2015/1/1 to 2015/1/20, the feature used in DPP is the 20-day candlestick chart built by the trading data between 2015/1/1 to 2015/1/20.

The parameters used in our study are set to be $k = 20$ and $m = 3$ in DPP. The assessment measure reported is the average of performance measures obtained from five randomly selected training and test data sets.

## Performance evaluation

In this section, we examine the performance of the proposed DPP based model, the index-based ensemble model IEM, the additive model Prophet, and the recurrent neural network based model LSTM. Each value reported in Tables 1 and 2 is the proportion of correct predictions made by a prediction model for the entire year.

**Evaluation on the TX dataset.**  To achieve the optimal results, we use data of all four futures in the TX dataset (i.e., TX, Mini-TX, TE and TF) to train DPP, IEM, and LSTM. With large number of parameters, deep learning models usually require more data in their training

**Table 1. Accuracy comparison for each year (TX dataset).**

| Year | Experiment | | | |
|---|---|---|---|---|
| | DPP | IEM | Prophet | LSTM |
| 2007 | 79.76% | 82.80% | 52.43% | 54.46% |
| 2008 | 82.28% | 84.94% | 48.79% | 61.94% |
| 2009 | 78.47% | 73.37% | 44.62% | 52.63% |
| 2010 | 77.46% | 78.75% | 51.79% | 54.82% |
| 2011 | 76.34% | 80.77% | 51.01% | 52.23% |
| 2012 | 84.09% | 82.06% | 57.62% | 55.94% |
| 2013 | 86.00% | 78.52% | 49.59% | 49.76% |
| 2014 | 79.68% | 77.49% | 56.45% | 58.22% |
| 2015 | 82.23% | 81.63% | 53.27% | 57.01% |
| 2016 | 81.86% | 79.42% | 52.04% | 56.56% |
| Overall Average (2007–2016) | 80.82% | 79.98% | 51.76% | 55.36% |
| Most Recent 5-Year Average (2012–2016) | 82.78% | 79.83% | 53.79% | 55.50% |

process to achieve satisfactory accuracy. Note that this is based on the assumption that different products with similar patterns in candlestick charts may lead to similar price movements. In contrast, only the TAIEX futures (i.e, TX, 4,625 records in total) are used to train Prophet since Prophet predicts the close price by extracting the characteristics of the time series of historical close prices. Table 1 shows the accuracy values of 4 models in each year of the TX dataset using nested cross-validation.

First, it is evident that Prophet and LSTM models are not comparable with DPP and IEM based models due to the relatively low accuracy values in Table 1. It can be seen that the accuracy values of DPP and IEM are comparable when data in the period from 2007 to 2016 are used, though DPP performs slightly better. However, DPP outperforms IEM by about 3% when the 2012–2016 data are used. An interesting finding here is that the accuracy of DPP in the recent five years (82.78%) is higher than that in the recent ten years (80.82%). On the contrary, the accuracy of IEM in the recent five years (79.83%) makes no significant difference compared to that in the recent ten years (79.98%). We find that DPP can achieve higher accuracy values than IEM in six out of ten years and this could imply that increase of the size in

**Table 2. Accuracy comparison for each year (NI225 dataset).**

| Year | Experiment | | | |
|---|---|---|---|---|
| | DPP | IEM | Prophet | LSTM |
| 2011 | 65.60% | 55.00% | 51.16% | 50.00% |
| 2012 | 66.37% | 52.77% | 38.63% | 57.33% |
| 2013 | 65.92% | 53.36% | 60.46% | 55.85% |
| 2014 | 65.90% | 54.38% | 55.81% | 54.75% |
| 2015 | 67.95% | 54.97% | 45.23% | 51.58% |
| 2016 | 66.36% | 55.00% | 54.76% | 54.05% |
| 2017 | 66.73% | 49.36% | 41.67% | 52.96% |
| 2018 | 67.51% | 57.45% | 50.00% | 53.15% |
| 2019 | 66.49% | 57.41% | 64.76% | 53.21% |
| Overall Average (2011–2019) | 66.53% | 54.41% | 51.39% | 53.65% |
| Most Recent 5-Year Average (2015–2019) | 67.08% | 54.84% | 51.28% | 52.99% |

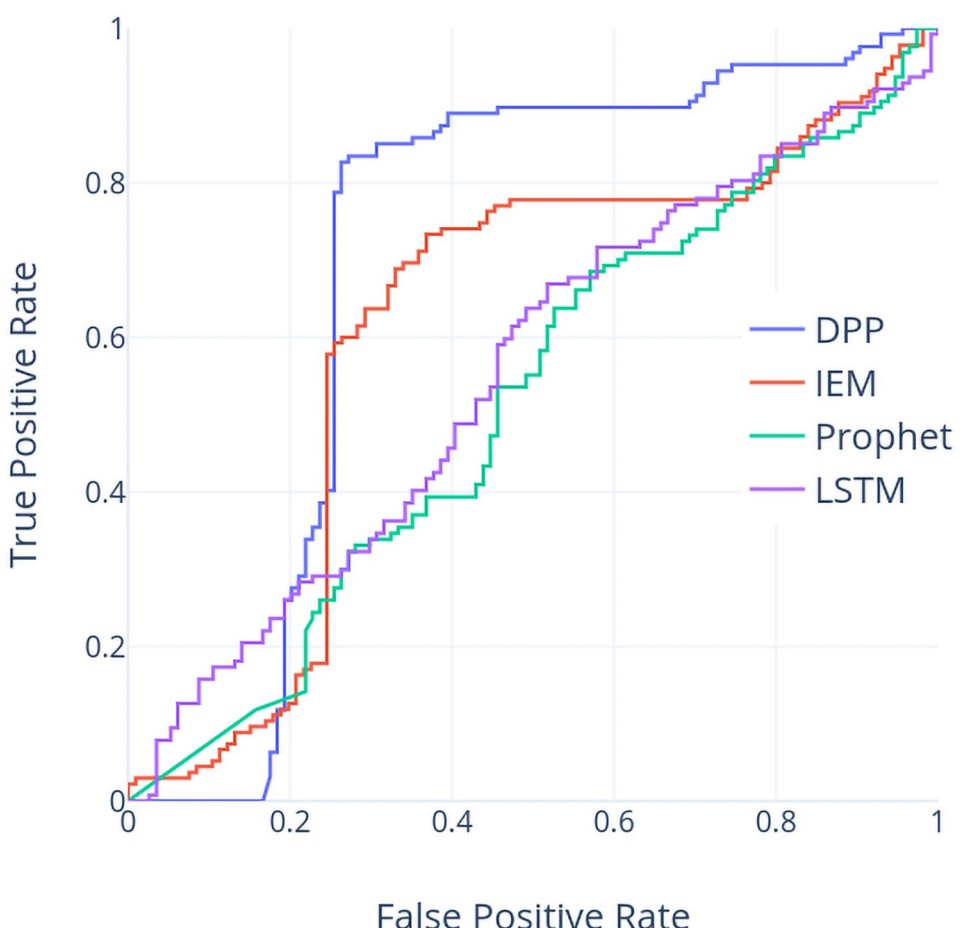

**Fig 9. ROC curve (TX dataset).**

training data could bring more advantage to DPP than IEM as more data could be used for model training in latter years in nested cross-validation. In general, deep-learning models have more parameters and require more data to tune these parameters to improve performance. As a deep-learning-based approach, DPP could achieve better accuracy with the increase of training data whereas IEM may not be able to do so in this experiment.

In addition, the receiver operating characteristic (ROC) curves and precision-recall (PR) curves of these four models based on the TX dataset are provided in Figs 9 and 10. Both figures show that the largest area under either ROC or PR curves is based on our proposed method DPP, which supports that the DPP based model is preferable compared to the other 3 models.

We can further compare the accuracy values over all approaches using the TX dataset by observing the accuracy report in Table 1 and the run chart in Fig 7. In year 2007, IEM outperforms all other approaches. The price in year 2007 has three big jumps and two big drops. Inferring the price trend by indices works well in this situation. In year 2008, IEM and DPP can predict the trend well when the price goes down with consecutive big drops. In year 2009 and year 2010, all accuracy values are relative lower compared to them in other years. The run chart shows that the price at the beginning of year 2009 is around the lowest price (about 4K) among the recent five years but the price keeps going up and the highest price is about 8K. The

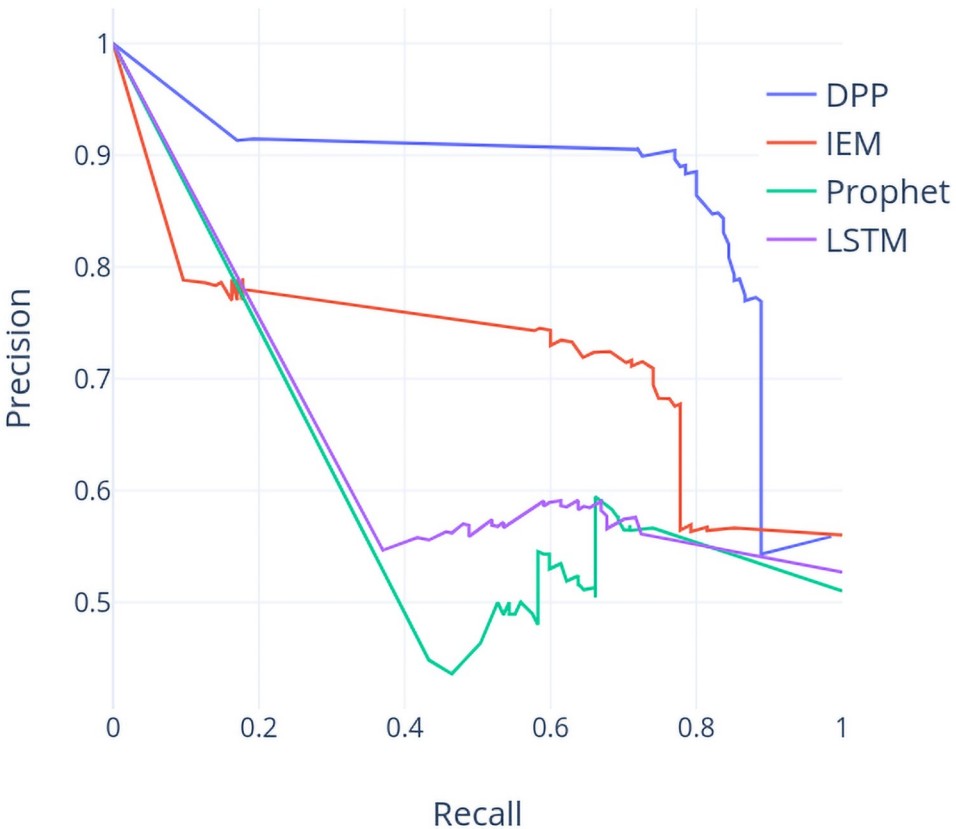

**Fig 10. PR curve (TX dataset).**

price movement is also choppy on mixed results and such a situation makes the prediction of the price movement more difficult for all approaches. Move to the results in year 2012 and year 2013, DPP reaches the highest accuracy whereas the accuracy of IEM drops suddenly in year 2013. We can observe that the price in year 2013 vibrates frequently although it presents a upward trend. This implies that DPP would outperform other approaches in such a situation.

**Evaluation on the NI225 dataset.** Since the NI225 dataset only includes open, close, high, and low prices, all models use all price measures in their training process. Table 2 shows the accuracy values of 4 models in each year of the NI225 dataset using nested cross-validation.

Contrast to the previous case using the TX dataset, it is clear that the DPP based models outperform the models using the other 3 approaches by about 10% or higher of the accuracy values. Figs 11 and 12 shows ROC and PR curves of these four models based on the NI225 dataset individually. Again, the largest area under either ROC or PR curves appears on the one based on DPP method, which indicates that DPP based model is superior to the other 3 models.

From the accuracy report in Table 2 and the run chart in Fig 8, we can observe that in year 2012, DPP can reach a relative high accuracy (66.37%) whereas IEM are with the lowest accuracy values (52.77% and 38.63%). A similar case is in year 2017 where the frequency and amplitude of price vibration are increasing over time. From both cases, DPP works well under the situation of frequent vibration of prices in the NI225 dataset whereas IEM suffers from this situation. The accuracy values for DPP are slightly lower in both year 2013 and year 2014 as the

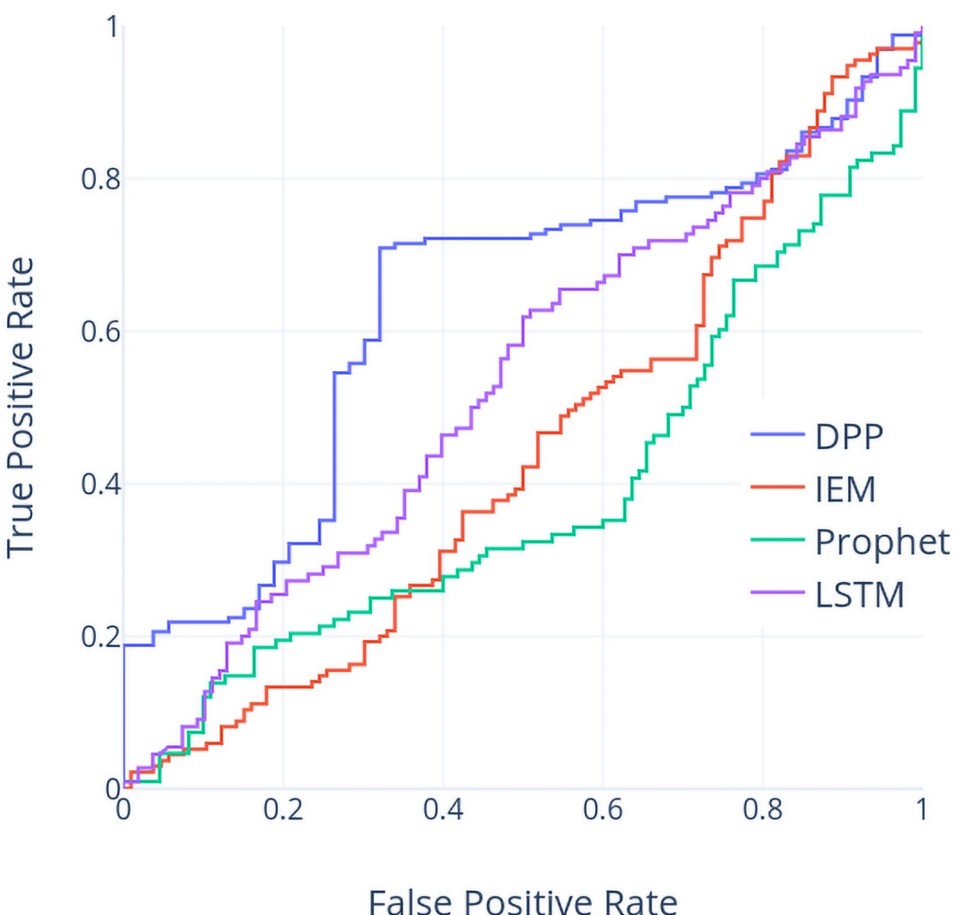

**Fig 11. ROC curve (NI225 dataset).**

price climbs from the lowest price among five years in the beginning of year 2013 to the relative highest price in the end of year 2014.

## Sensitivity analysis

We conduct the sensitivity analysis in this section. The results of models exposed to different time frames and data sources are firstly presented. The choice of hyper-parameters in DPP are then discussed.

This experiments start from how the accuracy of DPP and IEM in the TX dataset and the NI225 dataset changes in two different markets in the same time period. Here, we omit Prophet and LSTM since their accuracy values are not comparable to DPP and IEM. Since Taiwan and Japan are neighboring countries in Asia, they share a part of similar markets in their economic activities so that their state economies are usually correlated. Therefore, we can further identify the appropriate time when to use a model by such comparisons. We first select the price information from both datasets during 2011/1/1 to 2016/12/31. We then use Z-scores to normalize the prices for both datasets as shown in Fig 13. In year 2011, the prices in both datasets are down-trended. The amplitude of the TX dataset (1.62) is much larger than that of NI225 dataset (0.47). Moreover, there is a sudden drop in the middle of year 2001 in the TX

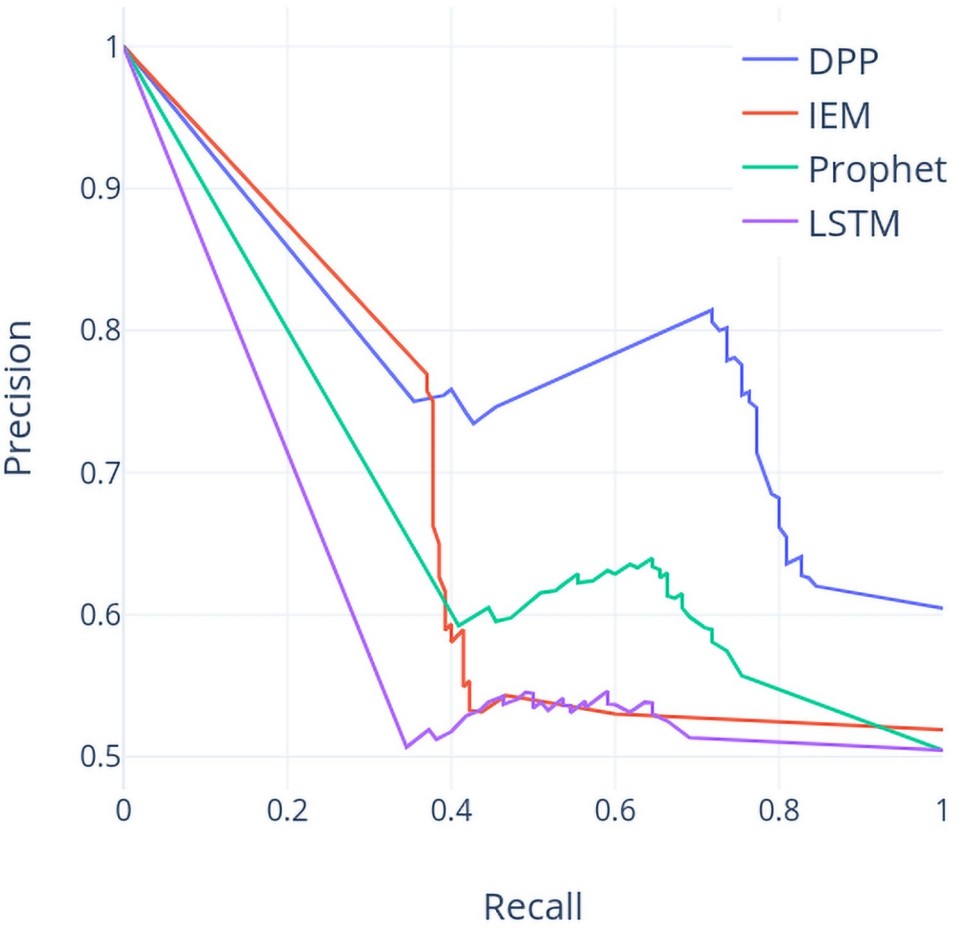

**Fig 12. PR curve (NI225 dataset).**

dataset. In this case, IEM can reach 3% higher accuracy than DPP. On the other hand, the price movement in the NI225 dataset is relatively stable. In this case, the accuracy of DPP is 10% higher than IEM. In year 2013, the price in the TX dataset goes up with many small bumps. In this case, DPP achieves the best accuracy (86.00%), which shows small bumps

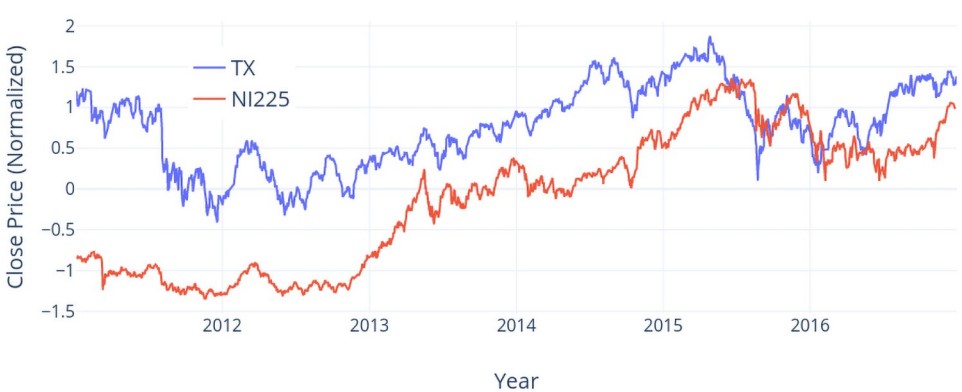

**Fig 13. The movement of close prices in TX dataset and NI225 dataset from 2011/1/1 to 2016/12/31.**

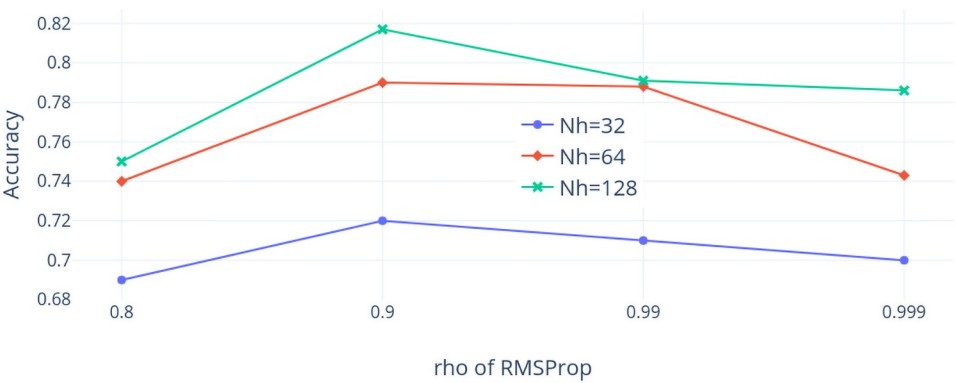

**Fig 14. Accuracy with $\rho$ varied (TX dataset).**

would not affect the prediction of DPP. On the other hand, the price in the NI225 dataset has a big jump in the first half of year 2013 and many bumps in the second half of this year. The accuracy of DPP (65.92%) and IEM (53.36%) in this year are relative lower than those in the other years. This shows the limitation of DPP and IEM.

This series of experiments explore how the accuracy of DPP varies as the hyper-parameters of the number of hidden features in GRU (denoted as $Nh$), the learning rate $\rho$ (denoted as rho) for RMSProp and the dropout rate are changed. Observing such evolution of hyper-parameters can help the selection of proper parameters to achieve the best accuracy using different datasets.

First, the TX dataset is considered. Fig 14 shows the accuracy values when the learning rate $\rho$ varied and the dropout rate is 0.5. The highest accuracy is around 81% when $\rho = 0.9$ and $Nh = 128$ and the accuracy seems to be sensitive to the change of $Nh$ but somehow less sensitive to the choice of $\rho$ when the dropout rate is 0.5. Fig 15 shows the accuracy values when the dropout rate varies and $\rho$ is 0.9. We find that the highest accuracy is around 81% when $Nh = 128$ and the accuracy is sensitive to the change of $Nh$ but less sensitive to the choice of the dropout rate when $\rho$ is 0.9. To conclude, the best combination of hyper-parameters among our choices using the TX dataset is that $\rho$ is 0.9, the dropout rate is 0.5, and $Nh$ is 128.

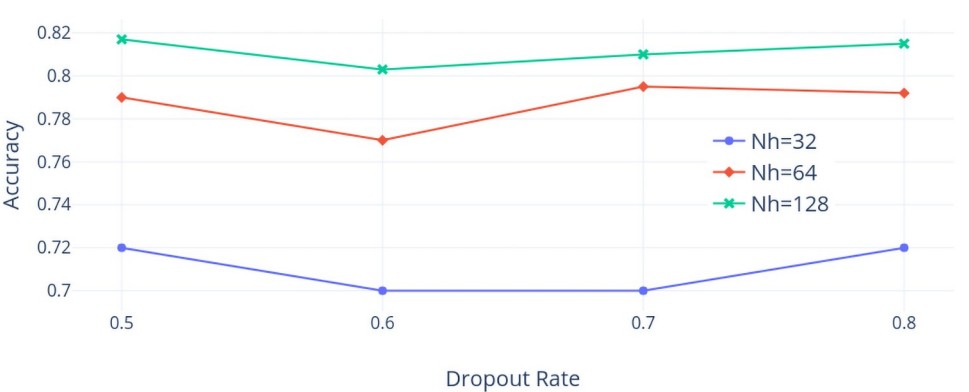

**Fig 15. Accuracy with droupout rate varied (TX dataset).**

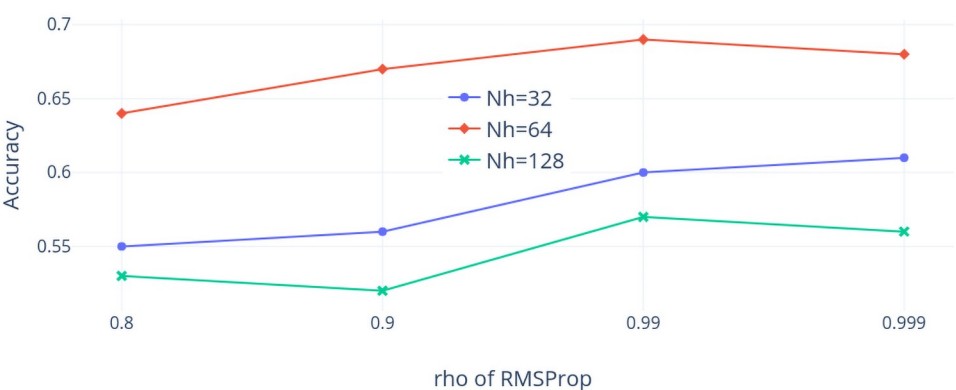

**Fig 16. Accuracy with $\rho$ varied (NI225 dataset).**

Second, the NI225 dataset is considered. Fig 16 shows the accuracy values as $\rho$ varies and the dropout rate is 0.7, while Fig 17 shows the accuracy as the dropout rate varies but $\rho$ is 0.99. We observe that the accuracy is sensitive to the choice of $Nh$ but slightly insensitive to the selection of $\rho$ when the dropout rate is 0.7. Similarly, the accuracy is again sensitive to the choice of $Nh$ and relatively insensitive to the selection of the dropout rate when $\rho$ is 0.99. In addition, the best combination of hyper-parameters among our choices using the NI225 dataset is that $\rho$ is 0.99, the dropout rate is 0.7, and $Nh$ is 64.

Note that the size of the NI225 dataset is around one-fourth of that of the TX dataset. The experimental results show that the optimal setting $Nh$ using the NI225 dataset is smaller than that using the TX dataset. A proper size of hidden features usually depends on the size of data. Learning a large size of hidden features from a small dataset usually suffers from the issue of over-fitting. We can also find that the optimal settings of $\rho$ and the learning rate using the NI225 dataset are larger than those using the TX dataset, respectively. A larger $\rho$ can help gradient to take more of the historical gradient than the coming gradient into account, which is helpful to find the optimal solution for a smaller dataset. A larger dropout rate is also a common way to prevent the over-fitting problem when the data size is relatively small.

## Conclusion

Technical analysis is used to predict price movements in the financial markets by using historical price charts and market statistics. Candlestick charts have been one of the most widely used charts as they contain many useful and explainable visual patterns for decision making. This

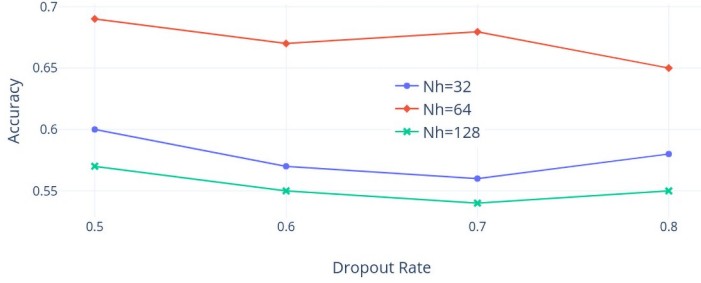

**Fig 17. Accuracy with droupout rate varied (NI225 dataset).**

paper proposed a framework DPP to predict the price movement by taking the candlestick charts as its input. Specifically, DPP firstly decomposes a given candlestick chart into several sub-charts, derives the best representation of sub-charts by a CNN autoencoder, and predicts the price movement by GRU. Extensive experiments are conducted using the Taiwan Exchange Capitalization Weighted Stock index and Nikkei 225, a stock index in Japan. The experimental results not only show that DPP outperforms the state-of-the-art methods but also discuss the effect of different stock markets to DPP.

## Supporting information

**S1 Data. TX dataset.** This dataset are from the Taiwan Futures Exchange (TAIFEX). The interval of this dataset is from July 21th, 1998 to December 27th, 2016, i.e., 4,625 trading days. (CSV)

## Author Contributions

**Writing – original draft:** Chih-Chieh Hung.

**Writing – review & editing:** Ying-Ju Chen.

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
