## [Decision Letter · Decision Letter 0]

5 Oct 2020

PONE-D-20-24642

DPP: Deep Predictor for Price Movement from Candlestick Charts

PLOS ONE

Dear Dr. Hung,

Thank you for submitting your manuscript to PLOS ONE. After careful consideration, I feel that it has merit but does not fully meet PLOS ONE’s publication criteria as it currently stands. Therefore, I invite you to submit a revised version of the manuscript that addresses the points raised during the review process.

I agree with the reviewer 2 that there are some weak points in this manuscript that seriously difficult the future publication. However, I have decided to give the authors the opportunity of revising the manuscript, but all concerns must be properly addressed. 

Why the authors consider that candlestick charts have the capacity of resuming all important information needed for prediction of price movement? Why the rest of the information can be considered random? How Candlestick charts have the ability of selecting the proper information?

By the other hand, model performance in different scenarios should be also presented as well as comparison with others state of the art.

I agree also with the reviewer that excess of performance could be considered as casual, so I consider that there are serious doubts that the conclusions are fully supported by the results of your experiment.

We look forward to receiving your revised manuscript.

Kind regards,

J E. Trinidad Segovia

Academic Editor

PLOS ONE

Journal Requirements:

3. Please note that PLOS ONE does not allow for the use of footnotes in its publications. As such, we ask you to remove all footnotes and move the information contained in them to the main text.

4. Thank you for submitting the above manuscript to PLOS ONE. During our internal evaluation of the manuscript, we found significant text overlap between your submission and the following previously published work, of which you are an author.

http://www.inderscience.com/offer.php?id=107821

Please revise the manuscript to rephrase the duplicated text, cite your sources, and provide details as to how the current manuscript advances on previous work. Please note that further consideration is dependent on the submission of a manuscript that addresses these concerns about the overlap in text with published work.

Reviewers' comments:

Reviewer's Responses to Questions

**Comments to the Author**

1. Is the manuscript technically sound, and do the data support the conclusions?

Reviewer #1: Yes

Reviewer #2: Yes

2. Has the statistical analysis been performed appropriately and rigorously? 

Reviewer #1: Yes

Reviewer #2: Yes

3. Have the authors made all data underlying the findings in their manuscript fully available?

Reviewer #1: No

Reviewer #2: Yes

4. Is the manuscript presented in an intelligible fashion and written in standard English?

Reviewer #1: Yes

Reviewer #2: Yes

5. Review Comments to the Author

Reviewer #1: 1) Please review more recent related work and discuss how your approach is better.

2) Explain in depth what deep learning architecture did you used.

3) Compare the end results with other related work.

Reviewer #2: Please, see the attached review saved and uploaded as "Review of the Manuscript PONE-D-20-24642.pdf"

Please, see the attached review saved and uploaded as "Review of the Manuscript PONE-D-20-24642.pdf"

6. PLOS authors have the option to publish the peer review history of their article (what does this mean?). If published, this will include your full peer review and any attached files.

Reviewer #1: **Yes: **Ashish Bhagchandani

Reviewer #2: No

---

## [Author Response · Author response to Decision Letter 0]

18 Apr 2021

We have already uploaded our response letters for editor and reviewers.

---

## [Decision Letter · Decision Letter 1]

17 May 2021

DPP: Deep Predictor for Price Movement from Candlestick Charts

PONE-D-20-24642R1

Dear Dr. Hung,

We’re pleased to inform you that your manuscript has been judged scientifically suitable for publication and will be formally accepted for publication once it meets all outstanding technical requirements.

Kind regards,

J E. Trinidad Segovia

Academic Editor

PLOS ONE

Additional Editor Comments (optional):

Reviewers' comments:

Reviewer's Responses to Questions

**Comments to the Author**

1. If the authors have adequately addressed your comments raised in a previous round of review and you feel that this manuscript is now acceptable for publication, you may indicate that here to bypass the “Comments to the Author” section, enter your conflict of interest statement in the “Confidential to Editor” section, and submit your "Accept" recommendation.

Reviewer #2: All comments have been addressed

2. Is the manuscript technically sound, and do the data support the conclusions?

Reviewer #2: Yes

3. Has the statistical analysis been performed appropriately and rigorously? 

Reviewer #2: Yes

4. Have the authors made all data underlying the findings in their manuscript fully available?

Reviewer #2: Yes

5. Is the manuscript presented in an intelligible fashion and written in standard English?

Reviewer #2: Yes

6. Review Comments to the Author

Reviewer #2: Thanks for the revision.

The editor will decide whether to accept or not the manuscript.

.

7. PLOS authors have the option to publish the peer review history of their article (what does this mean?). If published, this will include your full peer review and any attached files.

Reviewer #2: No

---

## [Editor Report · Acceptance letter]

3 Jun 2021

PONE-D-20-24642R1 

DPP: Deep Predictor for Price Movement from Candlestick Charts 

Dear Dr. Hung:

I'm pleased to inform you that your manuscript has been deemed suitable for publication in PLOS ONE. Congratulations! Your manuscript is now with our production department. 

Kind regards, 

on behalf of

Dr. J E. Trinidad Segovia 

Section Editor

PLOS ONE